# Shared and distinct pathways and networks genetically linked to coronary artery disease between human and mouse

Zeyneb Kurt[1,2†], Jenny Cheng[1,3†], Rio Barrere-Cain[1], Caden N McQuillen[1], Zara Saleem[1], Neil Hsu[1], Nuoya Jiang[1], Calvin Pan[4], Oscar Franzén[5], Simon Koplev[5], Susanna Wang[1], Johan Björkegren[5,6], Aldons J Lusis[4,7,8], Montgomery Blencowe[1,3*†], Xia Yang[1,3,9,10*]

[1]Department of Integrative Biology and Physiology, University of California, Los Angeles, Los Angeles, United States; [2]The Information School at the University of Sheffield, Sheffield, United Kingdom; [3]Interdepartmental Program of Molecular, Cellular and Integrative Physiology, University of California, Los Angeles, Los Angeles, United States; [4]Department of Medicine, Division of Cardiology, University of California, Los Angeles, Los Angeles, United States; [5]Department of Genetics & Genomic Sciences, Institute of Genomics and Multiscale Biology, Icahn School of Medicine at Mount Sinai, New York, United States; [6]Department of Medicine, (Huddinge), Karolinska Institutet, Huddinge, Sweden; [7]Departments of Human Genetics & Microbiology, Immunology, and Molecular Genetics, UCLA, Los Angeles, United States; [8]Cardiovascular Research Laboratory, David Geffen School of Medicine, UCLA, Los Angeles, United States; [9]Interdepartmental Program of Bioinformatics, University of California, Los Angeles, Los Angeles, United States; [10]Department of Molecular and Medical Pharmacology, University of California, Los Angeles, Los Angeles, United States

*For correspondence:
montyblencowe@g.ucla.edu
(MB);
xyang123@g.ucla.edu (XY)

†These authors contributed
equally to this work

Competing interest: The authors
declare that no competing
interests exist.

Reviewing Editor: Arya Mani,
Yale University, United States

**Abstract** Mouse models have been used extensively to study human coronary artery disease (CAD) or atherosclerosis and to test therapeutic targets. However, whether mouse and human share similar genetic factors and pathogenic mechanisms of atherosclerosis has not been thoroughly investigated in a data-driven manner. We conducted a cross-species comparison study to better understand atherosclerosis pathogenesis between species by leveraging multiomics data. Specifically, we compared genetically driven and thus CAD-causal gene networks and pathways, by using human GWAS of CAD from the CARDIoGRAMplusC4D consortium and mouse GWAS of atherosclerosis from the Hybrid Mouse Diversity Panel (HMDP) followed by integration with functional multiomics human (STARNET and GTEx) and mouse (HMDP) databases. We found that mouse and human shared >75% of CAD causal pathways. Based on network topology, we then predicted key regulatory genes for both the shared pathways and species-specific pathways, which were further validated through the use of single cell data and the latest CAD GWAS. In sum, our results should serve as a much-needed guidance for which human CAD-causal pathways can or cannot be further evaluated for novel CAD therapies using mouse models.

## eLife assessment

In this **important** study, the authors integrated genetic and genomic datasets from humans and mice to unveil shared networks and pathways associated with coronary artery disease. Their **compelling** analysis led to the identification of new regulatory genes and pathways in vascular tissues and in the liver, allowing for a more in-depth understanding of the pathogenesis of coronary artery disease.

## Introduction

Coronary artery disease (CAD) represents one of the leading causes of mortality worldwide (*Tsao et al., 2022*) and is the most common type of heart disease. CAD is primarily caused by atherosclerosis, or the buildup and hardening of plaque in the arteries, and can lead to arrhythmias, heart attack, and heart failure (*Momiyama et al., 2014*). It is a complex disease that involves numerous genetic and environmental factors, including poor diet, lack of exercise and smoking (*McCarthy et al., 2008*). An individual's risk of developing CAD may be reduced by lifestyle changes, medication, or surgery; however, our ability to mitigate CAD from the number one cause of death is still limited (*Khera et al., 2016*). Therefore, a more comprehensive understanding of CAD mechanisms will help develop new preventative and therapeutic strategies.

Mouse models have been extensively used to study atherosclerosis mechanisms and to test therapeutic drugs (*von Scheidt et al., 2017*; *Xiangdong et al., 2011*). There are many advantages to using mouse models, such as low maintenance cost, fast reproduction cycles, and the ability to control the environment and genetically manipulate their genomes. In addition, the mouse genome shares 95% of the protein coding genes with that of humans. However, there are important differences between the two species that make direct translation of findings from mice to humans less straightforward (*Vandamme, 2014*). For example, mice tend to develop larger atherosclerotic lesions in the aorta and carotids, whilst in humans the most consequential plaque lesions develop in coronary arteries (*Ma et al., 2012*). Because of the extensive use of mouse models in both mechanistic studies and preclinical investigations of therapeutic targets and drugs, a detailed understanding of the shared and distinct molecular mechanisms through a cross-species comparison will have tremendous translational value in cardiovascular research.

Previously, literature-based analyses revealed limited gene-level sharing between species but 70% sharing at the level of molecular pathways without considering the tissue context or gene regulatory networks (*von Scheidt et al., 2017*). As literature-based findings can be biased (*Stoeger et al., 2018*), here we conduct a comprehensive data-driven integrative study utilizing the ample multiomics data resources available to allow a tissue-specific, systems-level assessment of the key similarities and differences in CAD mechanisms between mouse and human. For each species, we focus on genetic and functional genomics datasets, including genome-wide association studies (GWAS) which uncover genetic risk loci, tissue-specific transcriptome data that reveal global gene expression and gene-gene coordination patterns, and tissue-specific expression quantitative trait loci (eQTLs) that reflect genetic regulation of gene expression. Instead of focusing on direct intersections of individual risk genes between species, our study builds on the 'omnigenic' disease model that emphasizes pathway and gene network perturbations to make mechanistic comparisons between species (*Boyle et al., 2017*). Indeed, recent studies support the involvement of tissue-specific pathways and networks in CAD and other complex diseases (*Blencowe et al., 2021*; *Chella Krishnan et al., 2018*; *Kurt et al., 2018*; *Zhao et al., 2019a*; *Zhao et al., 2019b*) and that cross-species comparison is more informative at the pathway level. Through multiomics integration, our study reveals both shared and species-specific pathways, networks, and key regulatory genes. Our findings will help inform on when mouse models are appropriate for testing different aspects of atherosclerosis and predict how well findings from mice will translate in human studies, ultimately increasing the likelihood of success in the design, development, and testing of new therapeutic agents for CAD.

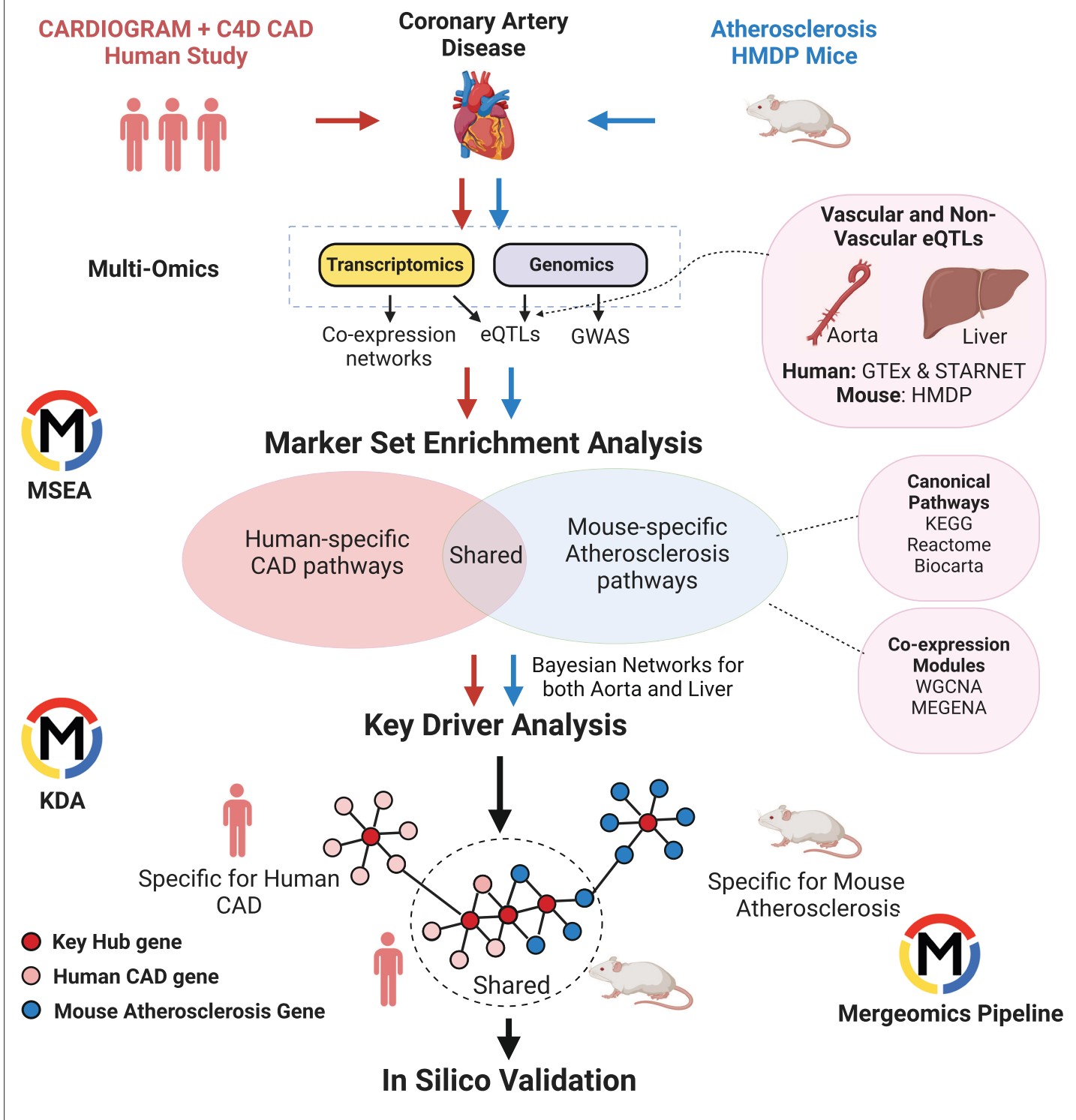

**Figure 1.** Overview of Study.

## Results

### Multiomics study design of CAD/Atherosclerosis

To understand the similarities and differences in CAD/atherosclerosis pathways and candidate disease regulatory genes in human and mouse, we used Mergeomics to integrate multiomics data including GWAS signals, tissue-specific eQTLs, and gene networks to predict causal CAD/atherosclerosis

subnetworks and their potential regulator genes in each species (*Figure 1*). From the human side, we included CAD GWAS from the Coronary Artery Disease Genome-wide Replication and Meta-Analysis (CARDIoGRAMplusC4D) consortium and transcriptomes and eQTLs from vascular tissues and liver from the GTEx consortium and the STARNET study. From the mouse side, we used atherosclerosis GWAS from the Hybrid Mouse Diversity Panel (HMDP), which is comprised of over 100 inbred strains of mice, with three to four mice per sex (*Bennett et al., 2015*), and transcriptomes and eQTLs from aorta and liver tissues from the same mouse cohort. We first constructed co-expression networks using gene expression data from vascular and liver tissues from both species and identified co-expression modules (clusters of co-expressed genes). We then mapped the human CAD GWAS and mouse atherosclerosis GWAS to genes using vascular and liver eQTLs. We used these mapped genes from GWAS to look for enrichment of gene sets (co-expression modules or canonical pathways), which will drive modules and pathways that are more likely to be causal and perturbed by genetic risk variants. We then identified gene sets that were shared and unique to each species and mapped the CAD/atherosclerosis-associated gene sets onto Bayesian networks constructed from liver and vascular tissues. These networks contain gene-gene regulatory relationships, which allowed us to predict candidate key driver (KD) genes in these CAD/atherosclerosis-associated gene sets that are potentially causal (see Methods for details).

## Construction of tissue-specific co-expression networks

We constructed co-expression networks from transcriptome data of 299 aortic artery, 173 coronary artery, and 175 liver tissue samples from the GTEx study (*Lonsdale et al., 2013*) for human, and 554 aortas and 508 liver samples of mice from 101 HMDP strains using two network methods, namely, Weighted Gene Co-expression Network Analysis (WGCNA; *Langfelder and Horvath, 2008*) and Multiscale Embedded Gene Co-expression Network Analysis (MEGENA; *Song and Zhang, 2015*). WGCNA and MEGENA cluster genes into modules based on the co-regulation structure of the genes using hierarchical clustering (details in Methods). Modules from both methods can be reciprocally conserved, however, their compactness and sizes are different, making these two methods complementary and allowing for different levels of biological pathways to be captured, as shown in our previous study (*Chella Krishnan et al., 2018*).

From the human datasets, we identified 36 co-expression modules in the aortic artery, 21 modules in the coronary artery, and 33 liver modules using WGCNA, whereas with MEGENA we obtained 159, 168, and 168 modules, respectively. For mouse, we identified 52 aorta co-expression modules and 55 liver modules using WGCNA, and 207 aorta and 190 liver modules using MEGENA (*Supplementary file 1A*). We pooled these co-expression modules along with 1823 canonical biological pathways from the Molecular Signatures Database (MSigDB) (*Subramanian et al., 2005*) to identify potential causal CAD/atherosclerosis processes in the next step using the Marker Set Enrichment Analysis (MSEA) function of Mergeomics. Here, co-expression modules represent functionally related genes defined by data-driven analysis, whereas MsigDB pathways capture functionally related genes through knowledge-driven categorization. They are complementary approaches to define genes sets containing functionally coherent genes, which can be combined to capture a potentially broader array of biology.

## Biological pathways and co-expression modules that exhibit genetic association with CAD/Atherosclerosis

Canonical biological pathways and co-expression modules from both species were pooled together, where each gene set (a pathway or module) contains functionally associated genes. We then mapped the GWAS data, which contains the potential causal signals, with the gene sets through species- and tissue-specific eQTLs to identify potential causal gene sets for CAD/atherosclerosis in each species and tissue. Instead of using only the top individual genome-wide significant signals in CAD GWAS, we used the whole spectrum of GWAS single nucleotide polymorphisms (SNPs) and their associated p-values, which allowed us to consider moderate and subtle signals in addition to strong ones. GWAS SNPs were then mapped to genes through species- and tissue-specific eQTLs which represent the functional association between genes and expression single-nucleotide polymorphisms (eSNPs) examined in GWAS. For each gene set among the pooled pathways and modules, we mapped the member genes to eSNPs through vascular (aorta in mouse, aorta and coronary arteries in human) or

non-vascular tissue (liver in both species) eQTLs by matching the tissue type between eQTLs and co-expression modules. Then, we used the Marker Set Enrichment Analysis (MSEA) procedure from our Mergeomics tool (*Ding et al., 2021*; *Shu et al., 2016*) to test whether an eSNP set, which was mapped from a given gene set, is enriched for stronger disease GWAS signals, based on GWAS p-value scores, compared to eSNP sets from random groups of genes (see Methods).

Among the 1823 canonical pathways and 1089 coexpression modules (643 from vascular tissue, and 446 from non-vascular tissue from both species; *Supplementary file 1A*), we found that human CAD GWAS signals from CARDIoGRAMplusC4D were significantly enriched in 47 pathways and 64 vascular tissue co-expression modules informed by vascular tissue eQTLs, whereas 59 pathways and 60 non-vascular (liver) modules were identified as CAD-associated by liver eQTLs (*Supplementary file 1B*). Mouse atherosclerosis GWAS signals from HMDP were significantly enriched in 68 pathways and 81 vascular tissue modules through the aortic eQTLs, and 37 pathways and 49 liver modules through liver eQTLs. Some of the CAD-associated pathways or modules share their member genes and correspond to similar biological processes, thereby creating redundancies. To reduce redundancy, we combined significantly overlapping gene sets (p<0.05, >33% gene overlap; see Methods) into 'supersets', yielding 73 and 80 supersets from human and mouse based on vascular tissue analysis, and 74 and 60 supersets from human and mouse based on liver analysis, respectively. Hence, each superset corresponds to one or more CAD/atherosclerosis-associated gene set (see Methods). These merged nonredundant supersets were found to retain enrichment for strong CAD or atherosclerosis-related genetic signals (*Supplementary file 1C*).

## Shared pathways and co-expression modules between human and mouse

We compared the CAD/atherosclerosis-associated supersets between mouse and human in each tissue separately. In addition to directly matching the pathways or modules by name or annotation, we also checked the overlap between the member genes of the supersets from both species. A two-sided Jaccard index overlap >50% or a one-sided overlap >90% was accepted as a match between species (see Methods). Since a superset that was found as CAD-associated in one of the species may cover >90% of the genes in more than one superset found in the other species, the number of the shared supersets from each of the species can differ.

Between vascular and liver tissues, we observed shared CAD pathways between tissues and species (*Figure 2*). These include well known signals in atherosclerosis development including metabolism of lipids and lipoproteins, extracellular matrix (ECM) organization, platelet activation/signaling/aggregation pathways, TCA cycle, and MAPK signaling.

In the vascular tissue-driven analysis, 53 of the 73 human supersets (73%) and 60 of the 80 mouse supersets (75%) were found to be matched or preserved between species (*Supplementary file 1C*, *Figure 2*). The shared pathways in vascular tissues include those less known for their role in the vasculature including gluconeogenesis, RXR/VDR pathway, and branched chain amino acid (BCAA) catabolism. At the same time, we uncovered some more common terms such as cell cycle, vascular smooth muscle contractions, and metabolism of lipids and lipoproteins (*Figure 2B*).

In the liver-based analysis, 56 of the 74 human supersets (76%) and 51 of the 60 mouse supersets (85%) were preserved between species (*Supplementary file 1C*, *Figure 2C*). The shared pathways in the liver include well characterized terms such as metabolism of lipids and lipoproteins fatty acid metabolism, Jak-STAT, PPAR signaling, circadian rhythm, and immune related signals (*Figure 2D*).

## Species-specific CAD/Atherosclerosis-associated mechanisms

We also identified species-specific processes in individual tissues (*Figure 2*). Human-specific pathways in vascular tissue analysis include a host of immune related signals (leukocyte transendothelial migration, viral myocarditis, and TLR-9 cascade) as well as intricate signaling cascades including GPCR and PDGF signaling, and more general terms such as diabetes pathway and metabolism pathways. Among these, leukocyte transendothelial migration and diabetes pathways were shared between species in liver (*Figure 2*, *Supplementary file 1C*). In liver, human-specific CAD pathways include immune system terms (interferon alpha/beta signaling, BCR signaling, and MHC class II antigen presentation), carbohydrate and BCAA metabolism, lysosome, neurotrophin signaling, and axon guidance

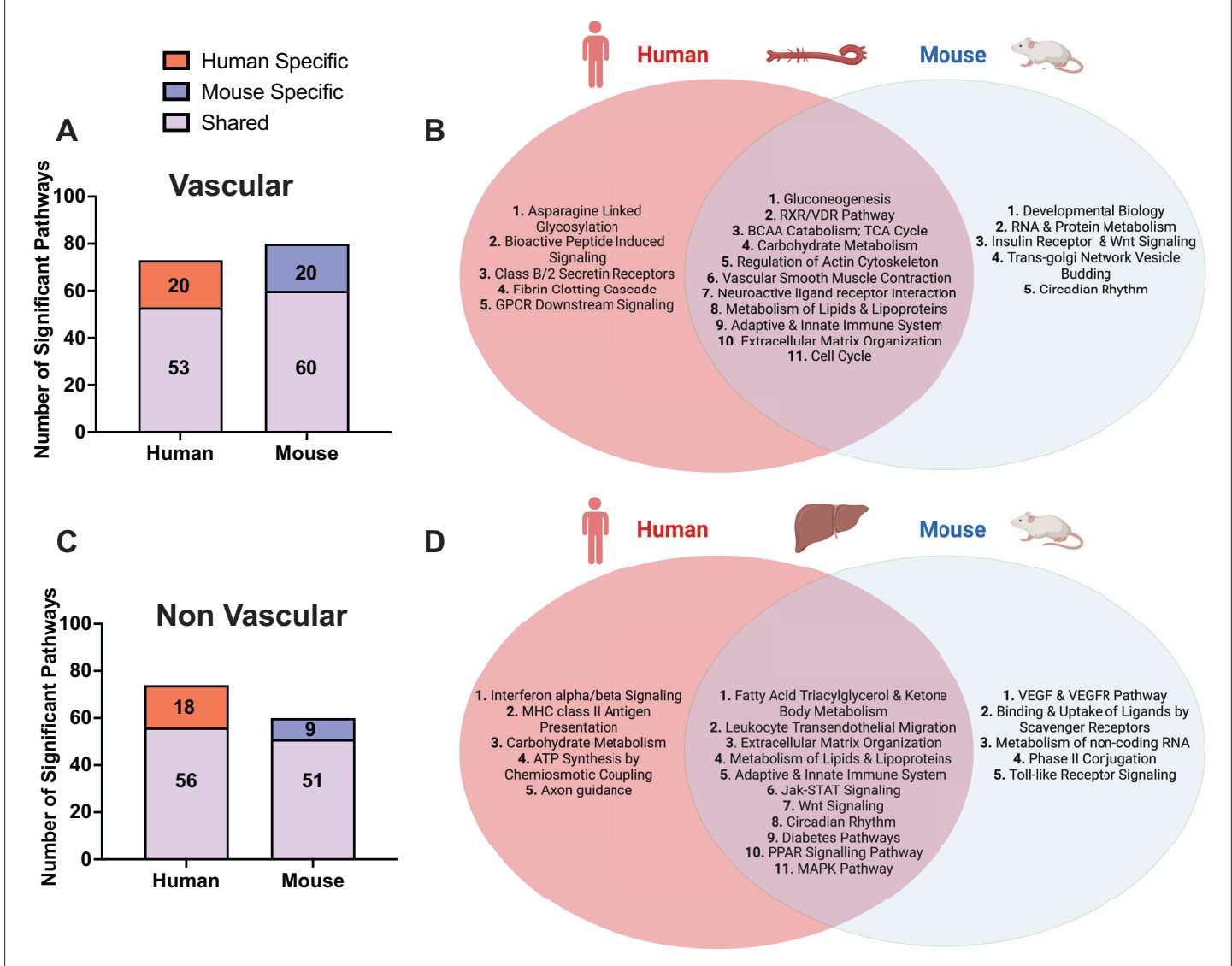

**Figure 2.** Shared and species-specific biological pathways. (**A**) Bar plot highlighting the number of significant pathways shared and unique between mouse and human in vascular tissue (FDR<0.05). (**B**) Venn diagram highlighting the top shared and unique significant pathways between mouse and human in vascular tissue. (**C**) Bar plot highlighting the number of significant pathways shared and unique between mouse and human in non-vascular tissue (liver) (FDR<0.05). (**D**) Venn diagram highlighting the top shared and unique significant pathways between mouse and human in non-vascular tissue (liver).

pathways, among which BCAA and carbohydrate metabolism pathways were found to be shared between species in the vascular tissue-based analysis.

Mouse-specific atherosclerosis mechanisms identified in the aorta include signaling by insulin receptor, EGFR, ERBB2, NGF, and TGF-beta signaling, axon guidance, VEGF and VEGFR pathways, and Tap63 and DeltaNp63 pathways (**Supplementary file 1B**). Some of the mouse-specific atherosclerosis pathways found in the aorta were identified as common to both species in liver tissue such as fatty acid synthesis, Jak-STAT and Wnt signaling. In liver tissue, mouse-specific pathways include toll-like receptor signaling, amino acid metabolism (arginine, proline, glycine, serine, and threonine), phospholipid metabolism, metabolism of non-coding RNA, binding and uptake of ligands by scavenger receptors, and mitochondrial transcription pathways (**Supplementary file 1B**). TLR signaling pathway was found to be mouse-specific in liver, whereas it was human-specific in vascular tissues. Similarly, pathways such as axon guidance and developmental biology were mouse-specific in aorta,

but human-specific in liver. Hence, some of the atherosclerosis pathways can be shared between species in a manner that is not matched by tissue-type.

## Identifying key driver (KD) genes in CAD/Atherosclerosis-associated supersets shared between mouse and human

We identified KDs, which are potential key regulatory genes, within the CAD/atherosclerosis-associated gene sets using the Key Driver Analysis (KDA) in the Mergeomics pipeline (*Ding et al., 2021*; *Shu et al., 2016*). The KDA procedure maps the tissue-specific CAD/atherosclerosis-associated gene sets onto a tissue-matched gene regulatory network to predict KDs (see Methods). Our analysis utilized Bayesian Networks (BNs), which incorporate gene expression patterns with genetic information and causal inference. Hence, BNs can reveal causal regulatory relationships between genes and enable the identification of potential regulators within the CAD/atherosclerosis-associated genes, canonical pathways or co-expression modules.

In vascular tissues, one of the top KDs for the shared CAD/atherosclerosis pathways between species is *ZHX2*, whose subnetwork neighbors are highly enriched for genes in a co-expression module annotated with RXR/VDR pathway, post translational protein modification, and endocytosis terms (*Figure 3A*). Other top ranked KDs include *MYLK, FLNA, ACTA2, NCAM1*, and *FOXC1*, which are associated with modules enriched for core matrisome, focal adhesion, and vascular smooth muscle contraction processes; *CEP350*, which is associated with a cell cycle, DNA replication, and B cell receptor activation module; *ASB5, MYF6, CACNA1S*, and *AMPD1* for a cardiac muscle contraction-associated module; *AGPAT1*, which is associated with a module annotated with metabolism of nucleotides and aminoacyl tRNA biosynthesis; *FLNA, CNN3*, and *MYL9*, which are associated with an ECM-related module; and lastly *MTA2*, which is associated with metabolism of carbohydrates, glycosaminoglycan degradation, and lysosome terms.

For liver tissue, the predicted KDs for the shared causal CAD/atherosclerosis pathways between mouse and human (*Figure 3B*) include: *NCKAP1L* and *INPP5D* for a Rho GTPase signaling module; *ARNTL* and *TEF* for circadian rhythm-related terms and pyrimidine metabolism modules; *COL1A1* and *COL6A3* for an ECM organization module; *RAC2* and *APBB1IP* for a platelet activation signaling and aggregation module; and lastly *SQLE* and *ACSS2* for a metabolism of lipids and lipoproteins module, which can be considered as a positive control pathway due to the critical role of circulating apoB lipoproteins in atherosclerosis and the central role of the liver in regulating their levels in both mice and humans.

Because KDA cannot predict directionality of the effects of KDs on CAD risk, we examined the correlation between KDs and atherosclerosis-relevant traits in the atherosclerosis HMDP as a means to suggest directionality. We found numerous significant positive or negative correlations for the vascular tissue KDs (*Supplementary file 1D*). For example, KDs *Agpat1* and *Zhx2* correlated positively with aortic lesion area, whereas *Cnn3* correlated negatively with aortic lesion area. Similarly, when querying the liver KDs in the atherosclerosis HMDP liver tissue, we found *Cidec* is negatively correlated with aortic lesion area as well as numerous KDs including *Inpp5d* and *Nckap1l* positively correlated with aortic lesion area (*Supplementary file 1E*).

## KDs for species-specific CAD/Atherosclerosis supersets

For human-specific vascular pathways, we identified *FNBP4, MYSM1, GC*, and *HPX* as the top KDs for a module related to fibrin clotting cascade and chylomicron mediated lipid transport; *PLCB2* and *SYK* for a leukocyte transendothelial migration module; and *LCK* and *CCL5* for a cell adhesion module (*Figure 4A*). For human-specific liver pathways, we identified *MCM5* and *MCM6* as KDs for a cell cycle module; *MX1, ISG15*, and *IRF7* for an interferon alpha/beta signaling module; *ACADM* for a module annotated with BCAA degradation; and lastly, *NNMT* for an IL6/7 signaling module (*Figure 4B*).

For mouse-specific vascular pathways, numerous KDs were identified: *Clec1a, Dll4, Myct1, Flt1*, and *Myo5c* for a VEGF/VEGFR pathway and axon guidance module; *Cep350, Malat1*, and *Utrn* for a fatty acid synthesis and ERBB2 signaling pathway module; *C2orf54, Ehf, Crabp2, Slc46a2*, and *Sox2* for a Tap63 and DeltaNp63 pathways module (*Figure 4C*). For the mouse-specific liver atherosclerosis pathways, *Ptprb, Kdr, Oit3*, and *Cyp4b1* were found to be the top KDs for a VEGF/VEGFR module; *Tmem43, Mtmr11, Plekha1*, and *Scnn1a* were the top KDs of a module related to amino acid

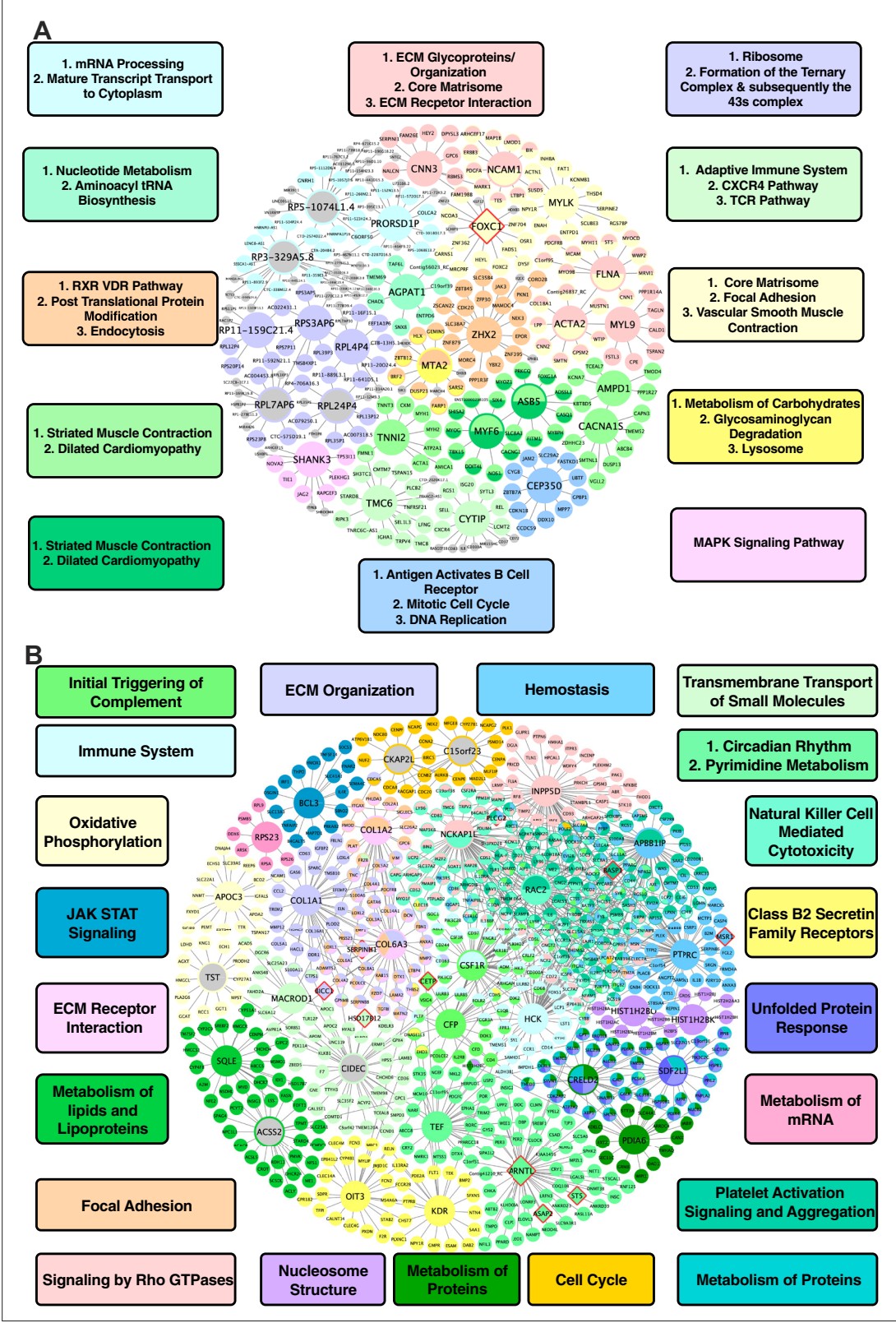

**Figure 3.** Shared networks between mice and humans. (**A**) Vascular tissue gene regulatory network shared between mice and humans (**B**) Non-vascular gene regulatory network shared between mice and humans. Each node is color coded based on the pathway/module that the genes are derived from with larger nodes signifying key driver genes. Red border diamonds represent CAD GWAS hits uncovered after the CARDIOGRAM+C4D GWAS (2016 onwards) and pink border diamonds represent CAD GWAS hits prior to the CARDIOGRAM+C4D GWAS.

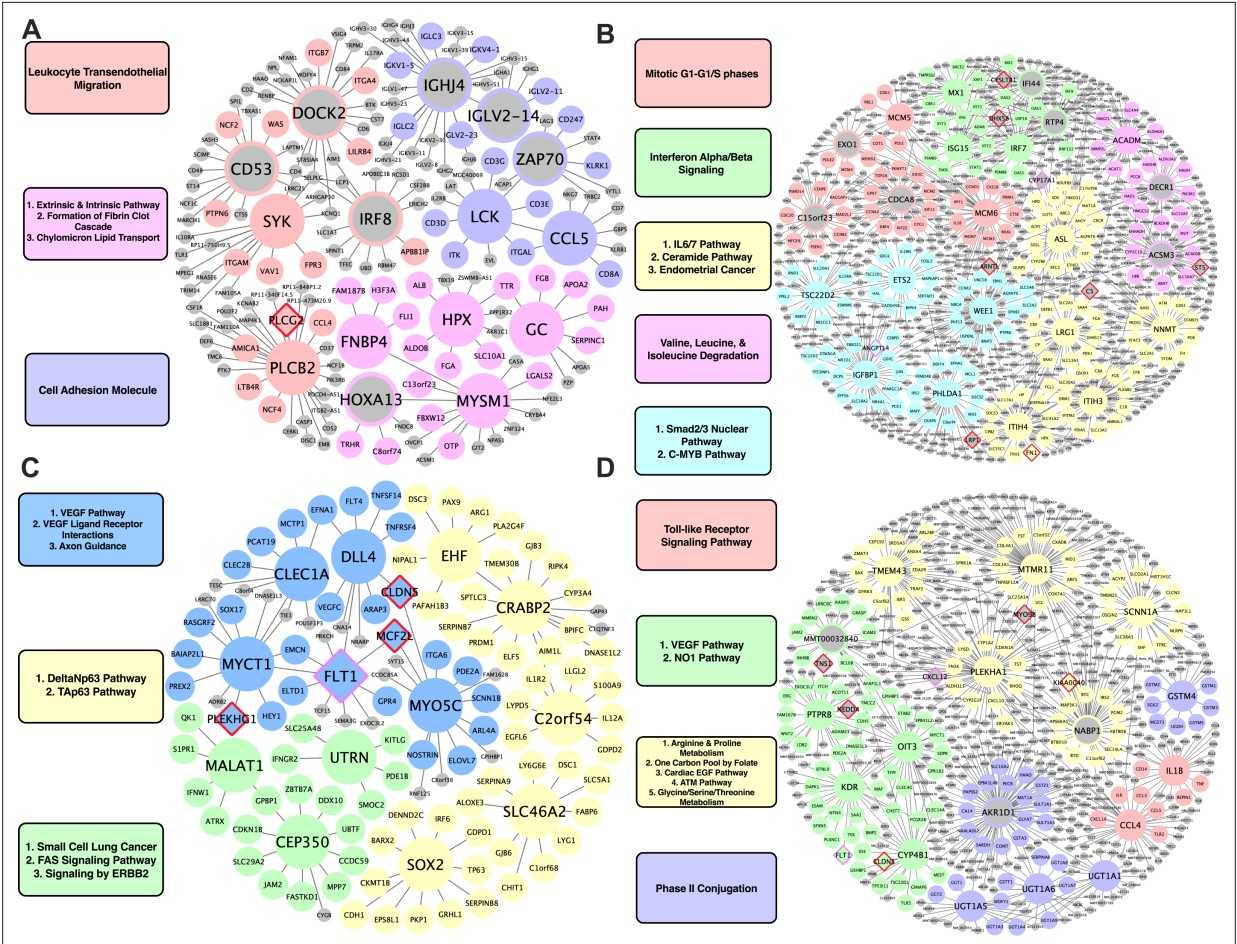

**Figure 4.** Species-specific networks. (**A**) Human vascular gene regulatory network, (**B**) Human non-vascular gene regulatory network, (**C**) Mouse vascular gene regulatory network, (**D**) Mouse non-vascular gene regulatory network. Each node is color coded based on the pathway/module that the genes are derived from with larger nodes signifying key driver genes. Red border diamonds represent CAD GWAS hits uncovered after the CARDIOGRAM+C4D GWAS (2016 onwards) and pink border diamonds represent CAD GWAS hits prior to the CARDIOGRAM+C4D GWAS.

metabolism (arginine and proline metabolism, glycine, serine, threonine metabolism) and cardiac EGF pathways (*Figure 4D*).

Examining the KD-trait correlations in HMDP showed many of these KDs are positively correlated (e.g. *Dock2*, *Irf8*, and *Zap70*) or negatively correlated (e.g. *Utrn*) with aortic lesion area in the atherosclerosis HMDP aorta tissue, or positively (e.g. *Ifi44*, *Irf7*, *Mx1*, *Nnmt*, and *Rtp4*) or negatively (e.g. *Acadm* and *Ets2*) correlated with aortic lesion area in the atherosclerosis HMDP liver (*Supplementary file 1D, E*).

## Comparison of KDs and pathways with known CAD/Atherosclerosis genes and processes in shared networks

Our analysis highlights a number of previously known CAD/atherosclerosis associated genes as KDs and commonly associated biological processes with CAD/atherosclerosis to be shared between species, but also uncovers those less known or less frequently associated to play a role in CAD development. For example, within vascular tissues (*Figure 3A*), we find a number of genes such as *NCAM1*, *MYLK*, and *ACTA2*, all of which have previously shown to have a link with CAD but with limited validation (*Erbilgin et al., 2018*; *Ghosh et al., 2015*; *Wang et al., 2007*; *Yuan, 2015*). Within the shared CAD/atherosclerosis liver network between species (*Figure 3B*), *APBB1IP*, *PTPRC*, *NCKAP1L* and *INPP5D* were captured as potential novel KDs which again have not been strongly investigated, validated or implicated in CAD previously. However, the associated biological pathways of these key driver genes, including Rho GTPase signaling, hemostasis, and platelet activation and aggregation

have all been repeatedly captured in CAD. Therefore, our analysis highlights novel regulators of these common CAD/atherosclerosis pathways.

## In silico validation of CAD/Atherosclerosis-associated networks through the use of single-cell transcriptome data

To validate the potential regulatory role of the KDs on their surrounding subnetwork genes and to pinpoint the potential cell types that they may be contributing through, we utilized mouse single-cell RNA-sequencing data from the aorta (*Wirka et al., 2019*) and liver tissues (*Wang et al., 2021*).

To validate our vascular tissue CAD/atherosclerosis networks (KDs and their direct edge connections), we compared *Apoe-/-* mouse aorta single-cell gene expressions at baseline condition and after the 8-week high-fat diet (HFD) treatment. We found that the subnetworks driven by the KDs *Acta2, Flna Mylk,* and *Myl9* exhibited significant changes in expression between diet conditions in smooth muscle cells (SMCs; *Figure 5A–D*), thus highlighting the validity of the network structure for these subnetworks and potentially their contribution to CAD development via SMCs. This is further supported in the literature, as *Acta2* mutations in SMCs have been linked to occlusive vascular disease via increased SMC proliferation (*Guo et al., 2009*). Our findings at the cellular level within SMCs suggest the neighboring subnetwork genes of *Acta2* cooperate with this KD to contribute to atherogenesis and CAD progression. Also, in regard to the KD *Flna*, interaction between G protein-coupled P2Y$_2$ nucleotide receptor (P2Y$_2$R) and FLNa, an actin-binding protein encoded by *Flna*, promotes SMC spreading, a hallmark characteristic in atherosclerosis development (*Yu et al., 2008*). Lastly, modification of Mlck, encoded by KD *Mylk* has been shown to disrupt the ability of aortic SMCs to generate force, leading to compromised contractile function (*Huang et al., 2018*), which may play an important role in CAD.

Due to the lack of single-cell datasets for the liver for CAD studies, we utilized single cell data for a common comorbidity of CAD, non-alcoholic steatohepatitis (NASH), which is an advanced stage of non-alcoholic fatty liver disease involving liver inflammation and damage and linked to obesity, insulin resistance, and CAD. We found that the KD *Oit3* and its predicted surrounding network genes showed significant changes in expression between healthy and NASH states across multiple cell types such as hepatic stellate cells, hepatocytes, mononuclear phagocytes, and T cells (*Figure 5E–H*).

To ensure the observed gene expression changes in the KDs and their subnetwork genes validated above were not random in single cell data, we chose five random genes and their subnetwork genes as negative controls to test their expression changes between control and disease states. Indeed, few of these randomly selected genes showed statistically significant changes.

## In silico validation of CAD/Atherosclerosis-associated KD networks through the use of the latest CAD GWAS

To test whether our CAD/atherosclerosis subnetworks predict new CAD GWAS loci, we overlaid hits from a more recent CAD GWAS (*Tcheandjieu et al., 2022*), which was not used in our analysis, as well as hits from prior CAD GWAS (p<5 × 10$^{-8}$) onto the shared CAD/atherosclerosis subnetworks between species. We found that KDs *FOXC1* in the shared vascular network and *ARNTL* in the shared liver CAD network were among GWAS hits. Additional GWAS hits in the shared cross-species CAD/atherosclerosis subnetworks were peripheral nodes (e.g. *CETP, PLCG2, BASP1, MSR1, ST5, ASAP2*; *Figure 3A, 3B*).

## Discussion

In order to understand the key similarities and differences between mouse and human in atherosclerosis/CAD pathogenesis, we carried out a cross-species comparison of the genetically driven atherosclerosis/CAD mechanisms between mouse and human by integrating multiomics data. We observed a high ratio of overlap between species in genetically driven pathways (~74% in aorta and ~80% in liver), which include established CAD processes (e.g. metabolism of lipid, lipoproteins and fatty acids), as well as more novel processes (e.g. asparagine N-linked glycosylation). Given the high ratio overlap between species for genetic pathways, we can highlight the importance of the in vivo mouse model for understanding human CAD. More importantly, our results shed light on when to utilize mouse models for translational benefit to humans by focusing on the shared pathways and networks, and

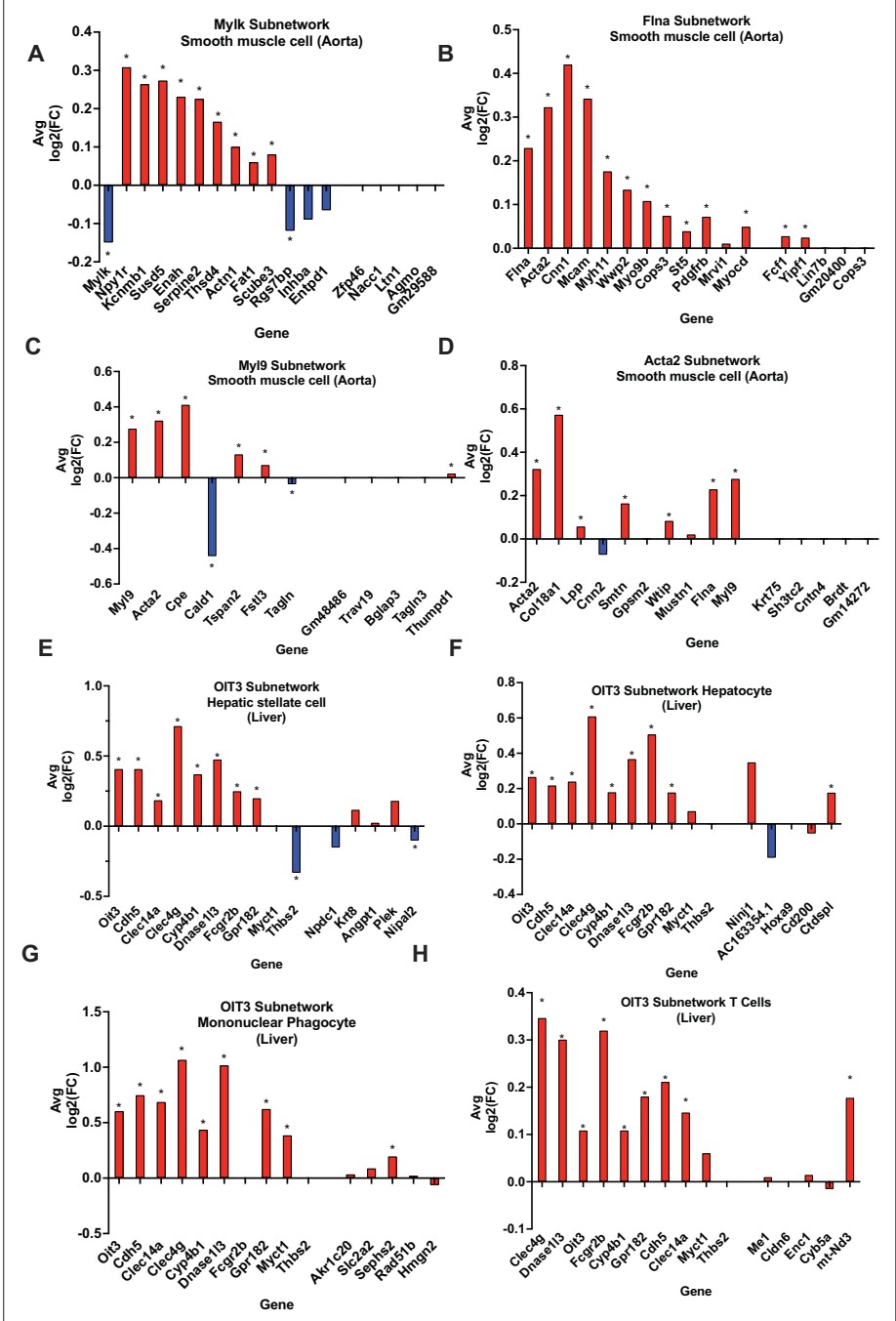

**Figure 5.** In silico validation of select KDs using single cell RNA-sequencing data. (**A-D**) Aorta KDs and subnetwork genes, where AvgLog2(FC) is representing gene expression change between atherogenic diet and chow diet groups. (**E-H**) Liver OIT3 KD and subnetwork genes, where AvgLog2(FC) is representing gene expression changes between NASH and control. The last 5 genes in each plot are randomly selected negative control genes. We utilized a Wilcoxon rank sum test with bonferroni correction to derive significance for each gene. * represents a p<0.05.

when we should look to alternative animal models for processes which are not shared between mouse and human (*Gisterå et al., 2022*).

Our data-driven multiomics analysis confirmed a number of causal pathways that were previously reported to be shared between species in a previous study based on literature review: platelet activation and signaling, metabolism of lipid and lipoproteins, cholesterol biosynthesis, cell cycle pathways,

focal adhesion, and Jak-STAT signaling (*von Scheidt et al., 2017*). The previous study focused on CAD genes curated from the literature without considering the tissue context; here, our data-driven approach uniquely indicates that these shared pathways are involved in both vascular and non-vascular tissues in both species. Additionally, unlike the earlier studies, we incorporated CAD GWAS, thereby providing genetic evidence on how these pathways relating to plaque initiation, buildup, and thrombosis genetically drive CAD/atherosclerosis in both species. In addition to the replicated pathways, our data-driven approach also uniquely uncovered ECM, mitochondria function (oxidative phosphorylation, TCA cycle, respiratory electron transport), MAPK signaling, membrane trafficking, and notch signaling pathways to be shared across tissues between species. Since these are shared between species, targeting them in mice may lead to similar results when studying cardiovascular diseases in humans.

Our tissue-specific network analysis also revealed interesting tissue-specificity in the shared cross-species pathways. For instance, unique vascular tissue pathways include apoptosis, vascular smooth muscle contractions, and RXR/VDR pathways, all of which have been previously linked to CAD. We observed that some of the causal CAD/atherosclerosis pathways shared between species in a single tissue were identified as species-specific in the other tissue. For instance, BCAA catabolism pathway was found to be shared between mouse and human in vascular tissue, whereas it was also identified as human-specific in liver. Similarly, the leukocyte transendothelial migration pathway was shared between species in liver, whereas it was human-specific in vascular tissues. We also identified species-specific pathways in each tissue individually. For instance, signaling by insulin receptor was found to be mouse-specific in aorta, but the diabetes pathway was impaired in both mouse and human liver. These results point to the importance of taking the tissue context into consideration when translating genes and pathways between mouse and human.

After identifying potential causal atherosclerosis/CAD pathways, we identified the top KD genes of the shared and species-specific pathways in each tissue. *ZHX2* was found to be one of the top KDs in an aorta coexpression module shared between mouse and human (*Figure 4A*). *ZHX2* was previously identified as a candidate gene for CAD, and its proposed mechanism was through modulating macrophage apoptosis (*Sen et al., 2014*). Since it was also shown to be a regulator of major genes behind lipoprotein metabolism (*Gargalovic et al., 2010*), it may also contribute to the development of atherosclerosis via lipid modulation. However, our network modeling indicates *ZHX2* has additional functions related to RXR/VDR, protein modification, and endocytosis in vascular tissue. Thus, future experimental testing is warranted to further explore the diverse mechanisms through which *ZHX2* modulates atherogenesis. In liver tissue, a notable KD for a cross-species module involved in immune response and platelet function includes *RAC2* (*Figure 3B*). Previously, *Rac2* was shown to prevent progressive calcification in experimental atherogenesis through the suppression of Rac1-dependent IL-1β, and in CAD patients it has been found that *RAC2* expression was decreased while IL-1β expression is increased (*Ceneri et al., 2017*).

Through our in silico single-cell analyses in mouse aorta and liver, we also highlight various other KDs to be taken into further consideration. In the mouse aorta, changes in expression levels of genes within subnetworks of KDs *Acta2, Flna, Myl9,* and *Mylk* proved significantly robust in SMCs between control and a pro-atherogenic condition (*Figure 5*). In the liver, the KD *Oit3* and its subnetwork genes showed robust changes in multiple cell types between NASH and control conditions. Previously, *Oit3* has been suggested to contribute to hepatic triglyceride homeostasis and very-low-density lipoprotein secretion, and hepatic OIT3 levels have been shown to be elevated in mice fed with HFD (*Wu et al., 2021*). Targeting *OIT3* may help treat both NASH and CAD. Importantly, the representation of our bulk aorta and liver tissue CAD networks within specific cell types helps provide support for the predicted gene networks interactions and warrants future in vivo cell type specific validation studies. In addition, through overlaying the latest human CAD GWAS hits onto our networks, we were able to highlight that our networks captured new GWAS hits such as KD *ARNTL* and peripheral network genes (e.g. *ASAP2, ST5, MCF2L, CLDN5, TNS1,* and *C5*) (*Figures 2–4*). Finding more GWAS hits to be peripheral nodes in the networks supports the notion that important regulators may not always harbor common variations due to evolutionary constraints. GWAS hits being found as peripheral hits is consistent with numerous prior studies (*Blencowe et al., 2021*; *Goh et al., 2007*; *Chella Krishnan et al., 2018*; *Zhao et al., 2019a*).

We acknowledge the following limitations of this study. First, we did not use the most recent and largest GWAS dataset as our initial input for our analysis. This potentially puts us at risk of not capturing the full array of biology. However, despite our input GWAS dataset not being the most recent, as mentioned above, we were able to capture *ARNTL* as a key driver gene connected to two more recent CAD GWAS hits *ASAP2* and *ST5* (*Aragam et al., 2022*; *Koyama et al., 2020*) and numerous other GWAS hits as network neighbors. Particularly, the identification of such genes acts as a form of validation of our network predictions and their importance in CAD pathogenesis. We also acknowledge that there are multiple approaches that can be used to map SNPs to genes beyond eQTLs such as splicing, protein, and epigenome QTLs (sQTLs, pQTLs, epiQTLs). Utilizing additional mapping methods as well integrating other omics layers such as metabolomics and plasma proteomics in the future will potentially highlight additional findings, which can complement our current tissue-specific gene network/ pathway results and possibly capture cross-tissue interactions mediated by secreted proteins and metabolites. In addition, it is important to note in this study that we did not regress out sex effects and we only examined HMDP as the mouse model of atherosclerosis. Therefore, there may be other pathways and genes that have been identified to be human specific in this study because they are missed in HMDP but can be captured in other mouse models, such as PDGF signaling found in the *Apoe-/-* model (*Kim et al., 2023*). Along with some pathways not being captured, we also highlight some interesting pathways being uncovered in vascular tissue, which have had little literature to support their role in atherosclerosis including gluconeogenesis and BCAA catabolism (*Zhenyukh et al., 2018*). We acknowledge that future studies will need to further investigate such pathway predictions but also note that these pathway terms have many shared genes with more commonly known processes such as the TCA cycle, which may be indicative of energy metabolism in the vasculature in CAD development. Lastly, we could not identify single cell datasets that matched the conditions of the human and mouse GWAS datasets used in our data-driven approach. Nevertheless, the network structures of the CAD KD genes appeared to be robust in conditions relevant to CAD, such as high fat diet feeding in *Apoe-/-* and NASH mice.

In summary, our comprehensive, data-driven, and integrative genomics study, highlights the similarities and differences in the molecular processes and key regulator genes involved in atherosclerosis and CAD between species. The incorporation of genetic information provides the potential causal nature of the pathways and regulators in CAD development. The novel insights highlighted here can enable the development of more effective therapies targeting the shared key regulator genes and pathways in light of their translatability from mouse to humans. More importantly, we are also made aware of the CAD human biology not captured in our mouse model, where other animal models should be considered. Both the shared processes between mouse and human and the human-specific processes warrant further investigation, and identifying non-mouse or additional mouse models that recapitulate the human-specific pathways is necessary for mechanistic and clinical translation.

## Methods

### Atherosclerosis study of Hybrid Mouse Diversity Panel

The Hybrid Mouse Diversity Panel (HMDP) study design for atherosclerosis was previously described (*Bennett et al., 2015*). Aortic lesion area phenotype was used as the hallmark of atherosclerosis. C57BL/6 mice carrying the human transgene for cholesteryl ester transfer protein were purchased from the Jackson Laboratory, and mice with the human apolipoprotein E-Leiden variant were obtained from Dr. Havekes (*van den Maagdenberg et al., 1993*). These strains were interbred to generate double transgenic mice, which were then bred to female mice from ~100 inbred or recombinant inbred strains obtained from the Jackson Laboratory. Female mice from the 100 strains were first fed with a chow diet for 8 weeks, and then placed on a high fat high cholesterol diet (33% kcal fat, 1% kcal cholesterol) until 24 weeks of age (Research Diets D10042101). Mice were then sacrificed for tissue collection. All experimental procedures were approved by the UCLA animal research committee.

### Atherosclerotic lesions in HMDP mice

Proximal aortic lesion area had been carried out as previously described (*Bennett et al., 2015*; *Mehrabian et al., 1993*; *Shih et al., 2000*). The aorta was cleaned with phosphate buffered solution and embedded in the optimal critical temperature compound. Oil Red O was used to stain frozen sections

of 10 μm. Lesion area was quantified in each third section across the proximal aorta. The aortic lesion size was normalized using the Yeo-Johnson approach (*Yeo and Johnson, 2000*).

## Gene expression analyses of aorta and liver tissues in HMDP

Tissue collection process for the expression data from aorta and liver tissues that were used in our analysis was previously described (*Bennett et al., 2015*). Whole aorta was cleaned of peri-adventitial adipose and immediately frozen at the time of euthanasia. The liver tissues were precisely dissected, and a 50 μg section from the left lobe was snap-frozen. Qiagen RNeasy kit was used to isolate total RNA for each tissue. Affymetrix HT-MG_430 PM microarrays were used to determine the genome-wide expression profiles for each tissue of female mice from 101 strains for aorta (1–10 aorta samples per strain, 366 samples in total) and 96 strains for liver (1–4 samples per strain, 220 samples in total) from the HMDP mice (*Bennett et al., 2015*). Additionally, we used gene expression data from an earlier HMDP study, which includes control mouse samples (*Bennett et al., 2010*). From these control mice, 188 aorta and 288 liver samples were collected (*Bennett et al., 2010*).

## Genome-wide association analysis of aortic lesion area and tissue-specific eQTLs in mice

Genotypes for 625 mice from 95 inbred and recombinant inbred mouse strains were obtained from Jackson Laboratories using the Mouse Diversity Array (*Yang et al., 2009*). SNPs with poor quality were removed (*Bennett et al., 2015*). Then, the SNPs with a minor allele frequency (MAF) of <10% and a missing genotype frequency of >10% were removed, resulting in 212,765 SNPs for the downstream analysis. Genome-wide association mapping of the aortic lesion size was performed using factored spectrally transformed linear mixed models (FaST-LMM) based on a linear mixed model to correct for population structure (*Bennett et al., 2015*). Tissue-specific eQTLs were also calculated using FaST-LMM and the false discovery rate (FDR) was estimated by the q value approach (*Storey and Tibshirani, 2003*) to adjust for multiple testing. In our analysis, we incorporated aorta tissue eQTLs from the atherosclerosis study for HMDP (Ath-HMDP) with aortic tissue eQTLs of another HMDP study, which included chow diet fed (control) HMDP mice that do not have atherosclerosis (*Bennett et al., 2010*). Similarly, we incorporated liver tissue eQTLs from multiple HMDP studies, including Ath-HMDP, control HMDP mice (*Bennett et al., 2010*), and high fat/high sucrose-fed HMDP mice with non-alcoholic fatty liver disease (NAFLD; *Parks et al., 2013*; *Parks et al., 2015*). In our study, cis-eQTLs, which are located within ±1 Mb of the genes' transcription start and end sites, and trans-eQTLs, which are identified as all other eQTL associations, were used with an FDR <5% cutoff for both. For each HMDP study, FDR scores of the cis- and trans-eQTLs were calculated independently (for the FDR <5%, e.g. p<6.4 × 1E-3 for Ath-HMDP aorta cis-eQTLs, 6.7×1E-3 for Ath-HMDP liver cis-eQTLs, p<1.3 × 1E-5 for Ath-HMDP trans-eQTLs for both aorta and liver [*Bennett et al., 2015*]; p<1.4 × 1E-3 and p<6.1 × 1E-6 for NAFLD-HMDP study cis-eQTLs and trans-eQTLs, respectively). Hence, we used a single, but stringent FDR <5% cutoff for all eQTL resources from multiple HMDP studies to provide consistent accuracy. In total, we used ~1.5 million eQTL associations in aorta tissue and ~3 million eQTL associations in liver.

## Human GWAS for coronary artery disease from the CARDIoGRAMplusC4D consortium

The coronary artery disease genome-wide replication and meta-analysis (CARDIoGRAM) and the coronary artery disease genetics (C4D) consortium (CARDIoGRAMplusC4D) collaborated and combined genetic data from diverse large-scale studies to determine risk loci for CAD and myocardial infarction (MI) in the human genome. In our analysis, we used the 1000 genomes-based GWAS from the CARDIoGRAMplusC4D, which is a meta-analysis of GWASs mainly based on European, South Asian, and East Asian descent, and imputed them through the 1000 Genomes phase 1 v3 training set with 38 million variants (*Nikpay et al., 2015*). In total, the GWAS includes 60,801 CAD cases, 123,504 control samples, and 9.4 million variants.

## Expression QTL data sets from the STARNET and GTEx

We used two different gene expression data resources: one from Stockholm-Tartu Atherosclerosis Reverse Network Engineering Task (STARNET) (*Franzén et al., 2016*), and the other from the

Genotype-Tissue Expression (GTEx) project (*Lonsdale et al., 2013*). We leveraged tissue-specific eQTLs from atherosclerotic aortic root (AOR), atherosclerotic-lesion-free internal mammary artery (MAM), and liver tissues from the STARNET cohort, which consists of ~600 CAD patients of Caucasian descent. Data collection and processing for STARNET were previously described (*Franzén et al., 2016*). AOR, MAM, and liver tissue samples were collected from 539, 552, and 576 individuals in total, respectively (*Franzén et al., 2016*). Written informed consent was obtained from all patients. Omni-Express Exome array was used to genotype the DNA and >14 million variant calls were identified (6,245,505 variants with a MAF >5%). HiSeq 2000 platform was used to perform RNA-sequencing, Matrix eQTL R-package (*Shabalin, 2012*) was used to identify the tissue-specific eQTLs, cis-eQTLs were defined within a±1 Mb window from the center of the gene, trans-eQTLs were identified as all other eQTL associations, cis-eQTL p-values were identified through a permutation test (for FDR <5%, p=0.006 for AOR, p=0.004 for MAM, p=0.001 for liver), and trans-eQTL p-values were corrected for multiple testing through the Bonferroni correction in *Franzén et al., 2016*. We used eQTL cutoff threshold FDR <5%.

GTEx consortium provides a publicly available human tissue bank with an enormous amount of data encompassing the associations between genotype and tissue-specific gene expression patterns in 54 tissues (*Lonsdale et al., 2013*). In our analysis, we used tissue-specific transcriptome and eQTL data from coronary artery, aortic artery, and liver tissues from the GTEx biobank as these were the most relevant tissues in CAD. Transcriptome data was used to construct tissue-specific coexpression networks, and eQTLs were used in our integrative framework, Mergeomics (*Shu et al., 2016*), to map gene sets to the CAD-GWAS SNPs (see MSEA in Methods). HiSeq X, HiSeq 2000, Illumina OMNI 5 M and 2.5 M arrays were used for genome sequencing. SNPs with a MAF ≥1% were kept for downstream analysis. Illumina TrueSeq RNA sequencing and Affymetrix Human Gene 1.1 ST Expression Array (v3) were used for gene expression data. As detailed in *Lonsdale et al., 2013*, RNA-sequencing data was aligned to the hg19/GRCh37 human reference genome using STAR v2.4.2a, based on the GENCODE-v19 annotation, and the GTEx pipeline was detailed in [47]. In the GTEx database, cis-eQTLs were identified within ±1 Mb of the genes' transcription start sites using the FastQTL tool (*Ongen et al., 2016*), which utilizes a linear regression model-based hypothesis testing. P-values were adjusted using the q-value method (*Storey and Tibshirani, 2003*) and significant eQTL associations with an FDR <5% threshold were used for each tissue. AOR and MAM tissue eQTLs from STARNET were combined with the vascular tissue (i.e. coronary and aortic arteries) eQTLs from GTEx. Similarly, liver tissue eQTLs from STARNET and GTEx were combined to use in our liver-specific analysis.

## Reconstruction of tissue-specific co-expression networks from liver and vascular tissue transcriptome data

Tissue-specific coexpression networks were constructed from aorta and liver gene expression data of the HMDP mice (*Bennett et al., 2015*; *Bennett et al., 2010*) and from aortic arteries, coronary arteries, and liver gene expression data from the GTEx biobank (*Lonsdale et al., 2013*), using two network approaches: Weighted Gene Co-expression Network Analysis (WGCNA; *Langfelder and Horvath, 2008*) and Multiscale Embedded Gene Co-expression Network Analysis (MEGENA; *Song and Zhang, 2015*). The power of WGCNA for identifying biologically meaningful and relevant modules has been demonstrated in multiple studies. However, the resulting large-sized modules which can include thousands of genes, may include noise and distract our focus in downstream analysis. To overcome this limitation, the other network method, MEGENA can define smaller and more coherent modules that are also biologically relevant. Additionally, each gene can be assigned into multiple modules by MEGENA, which overcomes another limitation of WGCNA and fits better with the known biology. In our study focusing on fatty liver disease, we showed these two methods complement each other while uncovering hidden parts of the biology that were missed by each method individually (*Chella Krishnan et al., 2018*). Both network methods are based on hierarchical clustering to assign co-regulated genes into the same coexpression module. Agglomerative hierarchical clustering is used in WGCNA, whereas divisive clustering is used in MEGENA. Gene-clusters are identified by merging (in agglomerative) or splitting (in divisive) based on a distance measure (e.g. 1-|correlation|). In WGCNA, 1 minus topological overlap matrix (TOM), hence dissTOM = 1 TOM, was used as the distance measure. TOM is based on the correlation score (edge weight) between two genes (nodes) but also considers the edge weights of common

neighbors of these two nodes in the network. To calculate the distance between two clusters, average dissTOM score of all gene pairs (each pair includes one gene from each cluster, while comparing 2 clusters) is used. In MEGENA, a shortest path distance (SPD) based distance measure is used. To create compact modules, a nested k-medoids clustering, which defines k-best clusters at each step that minimizes the SPD within each cluster, is used. Nested k-medoids clustering is ran until no more compact child cluster can be defined. Unlike WGCNA, MEGENA performs clustering in a multi-scale manner, which provides us an alternative set of modules at each scale despite using the same gene expression input. Multi-scale clustering assigns a gene into multiple modules from different scales.

After identifying the WGCNA and MEGENA modules, we annotated each module with its functions by using the previously curated biological pathways taken from MSigDB database (*Subramanian et al., 2005*) that incorporates pathways from Biocarta, KEGG, and Reactome databases based on a hypergeometric test (one-tailed Fisher Exact test). Bonferroni correction was used to correct for multiple testing. Pathways with a corrected p<5% and sharing ≥5 genes with a given module were considered significant. The top 5 significant pathways were used to annotate each module. For modules that did not have any significant annotation terms, less stringent cutoffs with a raw p<5E-3 and ≥5 shared genes were used. These modules are differentiated with an asterisk (*) in *Supplementary file 1*.

## Tissue-specific Bayesian network construction with RIMBANet

Bayesian networks (BNs) represent a class of gene regulatory networks, which can demonstrate the directed causal relationships between genes by using genetic and gene expression information, as well as previously known regulatory relationships between genes. In the BNs, each edge is directed to a child node from a parent node, where the BN represents a multivariate probability distribution, and the state of each node is estimated by the states of its parent nodes. We leveraged the Reconstructing Integrative Molecular Bayesian Networks (RIMBANet) tool (*Zhu et al., 2012*; *Zhu et al., 2007*; *Zhu et al., 2008*) to construct tissue-specific BNs using aorta and liver gene expression data from the HMDP study, focusing on atherosclerosis, which involved 366 aorta samples from 101 mouse strains and 220 liver samples from 96 strains (*Bennett et al., 2015*) as well as human GTEx aorta, coronary artery and liver gene expression data. Since the computational cost and computing time of RIMBANet are both high, we selected the top 5000 variable genes in the expression data to construct our networks. On average, it took ~2 weeks to construct a BN with 5000 genes. The expression values of each gene across all samples were discretized into 3 clusters as lowly, mildly, and highly-expressed using k-means clustering (*Alpaydin, 2010*). The state transitions of each node were formulated with Markov chain based on the discretized gene expression data. RIMBANet aims to find a network, which can explain the observed (gene expression) data in the best way and aims to maximize a joint probability function on the network nodes given the data. Given the expression data D, the joint probability of a network model M, that is P(M|D), can be predicted as a function of the prior probability of model M without any observations, that is P(M), and the likelihood of observing data D given the model M, that is P(D|M). Thereby, $P\left(M|D\right) \approx P\left(M\right) \times P\left(D|M\right)$. RIMBANet is based on Markov chain and because of the Markov equivalence concept, many of the edge directions in the network can be changed without affecting how well the model M fits the data D. There can be a large number of network models, which can explain the same observed data D equally well. Hence, it is necessary to narrow down the search-space while searching for the best-fitting network model. Using prior information based on the genetic data, eQTLs, tissue-specific transcription factor and target gene pairs can ensure narrowing the search space (e.g. a cis-acting gene is more likely to be a parent node of trans-acting genes coinciding on the same eSNP). Since it is not possible to calculate the joint probabilities of all possible network models given the expression data D, RIMBANet employs a down-sampling method, namely Monte Carlo Markov Chain (MCMC) simulation, identifies a certain number (1000) of plausible networks, and calculates the joint probabilities of these networks. Then, these 1000 sampled networks are combined to obtain a consensus network, for example if an edge from node-A to node-B exists in at least 30% of the 1000 networks, this edge is kept in the ultimate consensus network. BNs are directed acyclic graphs (DAG) by nature; thereby, the consensus network should also be a DAG, which is ensured by removing the weakest edges in a cycle that impairs the DAG nature of the network.

## Mergeomics pipeline for multi-omic integration

Genetic (atherosclerosis GWAS in mouse, CAD GWAS in human) and functional genomics data (eQTLs, coexpression modules) were integrated using our Mergeomics pipeline (*Arneson et al., 2016*; *Ding et al., 2021*; *Shu et al., 2016*). Canonical pathways and coexpression networks that are genetically associated with atherosclerosis GWAS (in mouse) or CAD GWAS (in human) were identified in a species- and tissue-specific manner by using the Marker Set Enrichment Analysis (MSEA) procedure in Mergeomics. MSEA is used to map genes from each pathway (from Biocarta, KEGG, Reactome) or coexpression module (from MEGENA or WGCNA) to an eSNP set via eQTLs of the corresponding tissue of the corresponding species. As mentioned earlier, we used significant eQTL associations (FDR <5%) from each tissue and species, and trimmed eSNPs within the same linkage disequilibrium (LD) block, by keeping only one eSNP for each LD block for each species individually. LD block data was obtained using the PLINK2 tool (*Chang et al., 2015*) for both HMDP mice and 1000Genome phase 1-based GWAS that was used in humans. The eSNP sets that were mapped from each pathway or module (i.e. gene set) through LD-pruned eQTLs, were annotated with the *P*-values of corresponding SNPs from the disease GWAS of corresponding species. Then, we used a modified chi-square statistic for the enrichment assessment, which is summarized across a range of quantile-based cutoffs for the GWAS, instead of depending on a single GWAS p-value cutoff. This test analyzes the significance of enrichment for stronger disease-GWAS p-values, by comparing the GWAS p-values of a given eSNP set against the eSNP sets that were mapped from randomly generated gene sets. Our MSEA approach is based on a set of quantile-based cutoffs and is not based on a single GWAS p-value cutoff; hence it avoids artefacts while producing more stable enrichment scores. The definition of the modified chi-square statistics used in the MSEA is: $\chi = \sum_{i=1}^{n} \frac{O_i - E_i}{\sqrt{E_i} + \kappa}$, where *O* and *E* are the numbers of the observed and estimated positive findings, respectively (i.e. findings above the *i*-th quantile point); n is the number of quantile points (10 points were identified ranging from the top 50% to top 99.9% signals based on the GWAS p-value rankings), and $\kappa = 1$ is a stability parameter diminishing the artefacts for small eSNP sets with low expected counts. For the MSEA procedure, an FDR <5% cutoff was used, which is estimated by the Benjamini-Hochberg method (*Benjamini and Hochberg, 1995*) to identify significantly enriched gene sets for the disease-GWASs.

After identifying the significant disease-associated gene sets (pathways or modules) at FDR <5%, to reduce the redundancy in our findings, we merged the overlapping gene sets into non-overlapping supersets if they significantly shared their member genes, which is defined by a gene-overlapping ratio of >33% and Bonferroni corrected one-tailed Fisher's exact test (FET) p<5%. In some cases, a coexpression module and a canonical pathway, or two different coexpression modules, which were annotated with the same biological term, were not merged since they did not significantly share their member genes; hence we kept them as independent gene sets though they were annotated with similar or the same terms.

The second step of the Mergeomics pipeline, Weighted Key Driver Analysis (wKDA), was used to predict key regulator, or key driver (KD), genes within the CAD-associated supersets. The wKDA maps the member genes in each superset onto a tissue-specific gene regulatory network, which is a Bayesian network (BN) in our study. wKDA has been demonstrated to predict biologically meaningful KDs in our previous studies (*Chella Krishnan et al., 2018*; *Mäkinen et al., 2014*; *Zhao et al., 2016*). In the current study, we first constructed aorta and liver tissue BNs, which include the causal and directed relationships between genes, using an established method RIMBANet (*Zhu et al., 2012*; *Zhu et al., 2007*; *Zhu et al., 2008*); then we mapped the CAD-associated supersets onto these BNs in a tissue-specific manner using wKDA to identify the KDs of each superset. wKDA uses a modified chi-square statistic, as defined for the MSEA, to assess the enrichment of member genes of a given superset within the candidate KD's neighborhood in the BN, compared to that of a random gene chosen from the same BN. Similar to MSEA, the Benjamini-Hochberg method was used to correct p-values for the multiple hypothesis testing, and candidate genes with an FDR <5% were predicted as KDs of the genetic CAD supersets. KDs of each shared and species-specific CAD superset were ranked based on their FDR scores and the top KDs were identified. Then, subnetworks of the top KDs were extracted in each tissue BN by collecting network-neighbors of the top KDs as illustrated in *Figures 4 and 5*. We overlaid CAD GWAS hits onto our subnetwork visualization where red border diamonds represent post 2016 CAD GWAS hits and pink border diamonds represent GWAS hits pre-CARDIOGRAM+C4D

GWAS. Lastly, we queried the atherosclerosis HMDP aorta and liver tissues for significant (p-value <0.05) gene-clinical trait correlations for shared, human-specific, and mouse-specific KDs.

## Comparing the genetic CAD pathways and modules between mouse and human

After identifying the putatively causal and genetic atherosclerosis and CAD-associated biological mechanisms using the MSEA, we compared our findings from mouse and human in a tissue-specific manner. To identify shared gene sets between species, we matched pathways and coexpression modules by name and annotation terms, as well as applied an overlap analysis to identify the common mechanisms. We calculated 2-sided (mutual) overlapping ratios between the CAD-associated mechanisms identified for mouse and human using the Jaccard index, $JI = \frac{|A \cap B|}{|A \cup B|} = \frac{n}{|A| + |B| - n}$, where n is the number of shared genes between supersets A and B, and |A| is the number of genes in set A. We also calculated 1-sided overlapping ratios as either $n/|A|$ or $n/|B|$ to identify gene sets that are potentially a subset of a larger set, and we assumed the subsets to be shared between species, whose member genes are already covered by both sides. Additionally, we applied a hypergeometric test (one-tailed FET) between the CAD-mechanisms identified for mouse and human. If a CAD superset from either mouse or human satisfied one of the following conditions when compared to a CAD superset identified for the other species, it was defined as a shared pathway:

1. A two-sided overlap that is >50% and a hypergeometric test adjusted p<5% or
2. A one-sided overlap that is >90% and a hypergeometric test adjusted p<5%.

Occasionally, multiple pathways in mouse were found to be a subset (one-sided overlap >90%) of a single pathway in human (or vice versa), or the two-sided overlapping analysis identified a pathway in one species to be matched to multiple pathways in the other species. Hence, the number of common pathways differs in each of the species as shown in *Figure 2* and *Supplementary file 1C*.

## In silico validation of network structure through use of single cell datasets

FASTQ files from each study (*Wirka et al., 2019*) for aorta, (*Wang et al., 2021*) for liver were aligned to the mouse or human reference genome using CellRanger Software from 10x Genomics. Each dataset was analyzed using the R package Seurat version 4.0.3. The datasets were filtered for cells expressing fewer than 200 genes or greater than 7500 genes to account for potential doublet events in which two separate cells are processed together. The number of genes, percentage of hemo-globin markers, and percentage of mitochondrial genes were observed to ensure quality control; cells containing >6% mitochondrial genes were considered of poor quality in liver and >15% for aorta and removed from the dataset. Gene expression values were normalized as feature counts for each cell were divided by the total counts for the cell and multiplied by a scale factor of 10,000; this value was then natural log transformed using log1p. Top genes differentially expressed in each cell type cluster were identified by FindAllMarkers in Seurat and subsequently used for further analysis of the datasets. Principal component analysis was performed for dimensionality reduction, and Uniform Manifold Approximation and Projection (UMAP) was then used for two-dimensional visualization of the clusters. To calculate the Average Log2 Fold Change in gene expression (AvgLog2FC) for the aorta, we compared mice on an atherogenic diet vs. chow. For the liver data, we compared NASH vs. control to calculate the AvgLog2FC. For both datasets we conducted a Wilcoxon rank sum test with bonferroni correction and considered a p<0.05 as significant. FASTQ files for aorta and for liver may be found in the Gene Expression Omnibus database respectively with the primary accession code GSE131780 and GSE166178.

## Acknowledgements

MB was supported by the AHA Predoctoral Fellowship (829009) and UCLA IBP Hyde Fellowship. ZK was supported by AHA Postdoctoral Fellowship (17POST33670739) and UCLA Iris Cantor, XY and AJL are supported by NIH R01 HL147883, DK117850. JB acknowledges support from the Swedish Research Council (2018–02529 and 2022–00734), the Swedish Heart Lung Foundation (2017–0265 and 2020–0207), the Leducq Foundation AteroGen (22CVD04) and PlaqOmics (18CVD02) consortia;

the National Institute of Health-National Heart Lung Blood Institute (NIH/NHLBI, R01HL164577; R01HL148167; R01HL148239, R01HL166428, and R01HL168174), American Heart Association Transformational Project Award 19TPA34910021, and from the CMD AMP fNIH program.

## Additional information

### Funding

| Funder | Grant reference number | Author |
|---|---|---|
| American Heart Association | AHA Predoctoral Fellowship (829009) | Montgomery Blencowe |
| University of California, Los Angeles | IBP Hyde Fellowship | Montgomery Blencowe |
| American Heart Association | AHA Postdoctoral Fellowship (17POST33670739) | Zeyneb Kurt |
| University of California, Los Angeles | Iris Cantor Center for Women's Health | Zeyneb Kurt |
| National Heart, Lung, and Blood Institute | R01 HL147883 | Aldons J Lusis Xia Yang |
| National Institute of Diabetes and Digestive and Kidney Diseases | R01 DK117850 | Aldons J Lusis Xia Yang |
| Swedish Research Council | 2018-02529 | Johan Björkegren |
| Swedish Research Council | 2022-00734 | Johan Björkegren |
| Swedish Heart-Lung Foundation | 2017-0265 | Johan Björkegren |
| Swedish Heart-Lung Foundation | 2020-0207 | Johan Björkegren |
| Leducq Foundation | AteroGen 22CVD04 | Johan Björkegren |
| PlaqOmics | 18CVD02 | Johan Björkegren |
| National Heart, Lung, and Blood Institute | R01HL164577 | Johan Björkegren |
| National Heart, Lung, and Blood Institute | R01HL148167 | Johan Björkegren |
| National Heart, Lung, and Blood Institute | R01HL148239 | Johan Björkegren |
| National Heart, Lung, and Blood Institute | R01HL166428 | Johan Björkegren |
| National Heart, Lung, and Blood Institute | R01HL168174 | Johan Björkegren |
| American Heart Association | Transformational Project Award 19TPA34910021 | Johan Björkegren |

The funders had no role in study design, data collection and interpretation, or the decision to submit the work for publication.

### Author contributions

Zeyneb Kurt, Data curation, Funding acquisition, Resources, Supervision, Writing – review and editing; Jenny Cheng, Data curation, Formal analysis, Investigation, Methodology, Validation, Visualization, Writing – original draft, Writing – review and editing; Rio Barrere-Cain, Funding acquisition, Resources, Writing – review and editing; Caden N McQuillen, Data curation, Funding acquisition, Investigation, Methodology; Zara Saleem, Funding acquisition, Investigation, Methodology; Neil Hsu, Investigation,

Methodology; Nuoya Jiang, Funding acquisition, Methodology; Calvin Pan, Oscar Franzén, Simon Koplev, Writing – original draft, Data curation; Susanna Wang, Methodology; Johan Björkegren, Aldons J Lusis, Writing – original draft, Data curation, Validation, Visualization, Methodology; Montgomery Blencowe, Conceptualization, Data curation, Formal analysis, Investigation, Methodology, Supervision, Visualization, Writing – original draft, Writing – review and editing; Xia Yang, Conceptualization, Data curation, Funding acquisition, Investigation, Methodology, Project administration, Resources, Supervision, Validation, Writing – original draft, Writing – review and editing

### Author ORCIDs

Zeyneb Kurt ⓘ http://orcid.org/0000-0003-3186-8091
Jenny Cheng ⓘ http://orcid.org/0009-0009-2248-1697
Caden N McQuillen ⓘ http://orcid.org/0000-0002-7762-9283
Oscar Franzén ⓘ http://orcid.org/0000-0002-7573-0812
Susanna Wang ⓘ http://orcid.org/0009-0006-0254-8418
Aldons J Lusis ⓘ http://orcid.org/0000-0001-9013-0228
Montgomery Blencowe ⓘ https://orcid.org/0000-0001-7147-1895
Xia Yang ⓘ https://orcid.org/0000-0002-3971-038X

Reviewer #1 (Public Review): https://doi.org/10.7554/eLife.88266.3.sa1
Reviewer #2 (Public Review): https://doi.org/10.7554/eLife.88266.3.sa2
Author Response https://doi.org/10.7554/eLife.88266.3.sa3

## Additional files

### Supplementary files

• Supplementary file 1. Co-expression module numbers, MSEA pathway results and KD correlations with CAD clinical traits. (A) The number of modules derived from WGCNA and MEGENA from vascular and non-vascular tissues. (B) Marker Set Enrichment Analysis (MSEA) results with an FDR <5% for Human CAD GWAS and Mouse atherosclerosis GWAS in vascular and non-vascular tissue. (C) Second round of MSEA results for the independent supersets with an FDR <5% and cross-species comparison. (D) Key driver correlations in vascular tissue with clinical traits relevant to CAD. (E) Key driver correlations in non-vascular tissue with clinical traits relevant to CAD.

• MDAR checklist

### Data availability

This a computational study, so no data has been generated for this manuscript. Any datasets utilized are cited within the methods section. For the analysis conducted we utilized an open access R code package called Mergeomics, which can be found using the URL: https://bioconductor.org/packages/release/bioc/html/Mergeomics.html (*Makinen et al., 2023*).

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
