## [Editor Report · eLife assessment]

In this **important** study, the authors integrated genetic and genomic datasets from humans and mice to unveil shared networks and pathways associated with coronary artery disease. Their **compelling** analysis led to the identification of new regulatory genes and pathways in vascular tissues and in the liver, allowing for a more in-depth understanding of the pathogenesis of coronary artery disease.

---

## [Referee Report · Reviewer #1 (Public Review)]

This manuscript represents an elegant bioinformatics approach to addressing causal pathways in vascular and liver tissue related to atherosclerosis/coronary artery disease, including those shared by humans and mice and those that are specific to only one of these species. The authors constructed co-expression networks using bulk transcriptome data from human (aorta, coronary) and mouse (aorta) vascular and liver tissue. They mapped human CAD GWAS data onto these modules, mapped GWAS SNPs to putatively causal genes, identified pathways and modules enriched in CAD GWAS hits, assessed those shared between vascular and liver tissues and between humans and mice, determined key driver genes in CAD-associated supersets, and used mouse single-cell transcriptome data to infer the roles of specific vascular and liver cell types. The overall approach used by the authors is rigorous and provides new insights into potentially causal pathways in vascular tissue and liver involved in atherosclerosis/CAD that are shared between humans and mice as well as those that are species-specific. This approach could be applied to a variety of other common complex conditions.

---

## [Referee Report · Reviewer #2 (Public Review)]

Summary:

Mouse models are widely used to determine key molecular mechanisms of atherosclerosis, the underlying pathology that leads to coronary artery disease. The authors use various systems biology approaches, namely co-expression and Bayesian Network analysis, as well as key driver analysis, to identify co-regulated genes and pathways involved in human and mouse atherosclerosis in artery and liver tissues. They identify species-specific and tissue-specific pathways enriched for the genetic association signals obtained in genome-wide association studies of human and mouse cohorts.

Strengths:

The manuscript is well executed with appropriate analysis methods. It also provides a compelling series of results regarding mouse and human atherosclerosis.

---

## [Author Response]

The following is the authors’ response to the original reviews.

**Reviewer #1 (Public Review):**
This manuscript represents an elegant bioinformatics approach to addressing causal pathways in vascular and liver tissue related to atherosclerosis/coronary artery disease, including those shared by humans and mice and those that are specific to only one of these species. The authors constructed co-expression networks using bulk transcriptome data from human (aorta, coronary) and mouse (aorta) vascular and liver tissue. They mapped human CAD GWAS data onto these modules, mapped GWAS SNPs to putatively causal genes, identified pathways and modules enriched in CAD GWAS hits, assessed those shared between vascular and liver tissues and between humans and mice, determined key driver genes in CAD-associated supersets, and used mouse single-cell transcriptome data to infer the roles of specific vascular and liver cell types. The overall approach used by the authors is rigorous and provides new insights into potentially causal pathways in vascular tissue and liver involved in atherosclerosis/CAD that are shared between humans and mice as well as those that are species-specific. This approach could be applied to a variety of other common complex conditions.The conclusions are largely supported by the analyses. Some specific comments:1. It appears that GWAS SNPs were mapped to genes solely through the use of eQTLs.Current methods involve a number of other complementary approaches to map GWAS SNPs to effector genes/transcripts and there is the thought that eQTLs may not necessarily be the best way to map causal genes.

We agree with the reviewer that multiple approaches can be used to map GWAS SNPs to genes, and eQTLs is only one way to do so. We focused on eQTLs mainly because we aim to address tissue-specificity of eQTLs and the relative higher abundance of eQTLs compared to other tissue-specific functional genomics data, such as pQTLs and epiQTLs. We now acknowledge this limitation in the discussion section in our revised manuscript and point to future studies utilizing other approaches to map GWAS signals to downstream effectors.

1. Given the critical causal role of circulating apoB lipoproteins in atherosclerosis in both mice and humans and the central role of the liver in regulating their levels, perhaps the authors could use the 'metabolism of lipids and lipoproteins' network in Fig 3B as a kind of 'positive control' to illustrate the overlap between mice and humans and the driver genes for this network.

We appreciate the reviewer’s excellent suggestion and now elaborate the findings in Fig 3B as a positive control in the results section.

1. Is it possible to infer the directionality of effect of key driver genes and pathways from these analyses? For example, ACADM was found to be a KD gene for a human-specific liver CAD superset pathway involving BCAA degradation. Are the authors able to determine or predict the effect of genetically increased expression of ACADM on BCAA metabolism and on CAD? Or the directionality of the effect of the hepatic KD gene OIT3 on hepatic and plasma lipids and atherosclerosis.

The Bayesian networks only have information on which genes likely regulate the others but not the up or down-regulation direction, and the network key driver analysis only considers the enrichment of GWAS candidate genes in the neighborhood of each key driver. Therefore, it is not possible to directly infer whether increasing or decreasing a key driver will lead to up or downregulation of the downstream pathways based on our current analysis. We could, however, examine correlations of key driver genes with downstream genes, or disease traits in relevant datasets. For instance, we checked the mouse atherosclerosis HMDP datasets for the correlations between select key drivers and clinical traits and found various key drivers shared and species-specific in aorta and liver significantly correlate with aortic lesion area and other traits of interest such as LDL levels, and inflammatory cytokines. We have added these new findings into the results section and supplemental tables.

1. While likely beyond the scope of this manuscript, the substantial amount of publicly available plasma proteomic and metabolomic data could be incorporated into this multiomic bioinformatic analysis. Many of the pathways involve secreted proteins or metabolites that would further inform the biology and the understanding of how these pathways relate to atherosclerosis.

We appreciate the reviewer’s valuable suggestion. Here we focused on liver and aorta gene regulatory networks to understand the tissue-specific mechanisms at the gene level. Indeed, plasma proteomic and metabolomic data could be further incorporated in future studies to understand the pathways captured in the circulation that can capture cross-tissue interactions mediated by secreted proteins and metabolites from different tissues. We have addressed this as a future direction in the discussion section.

The findings here will motivate the community of atherosclerosis investigators to pursue additional potential causal genes and pathways through computational and experimental approaches. It will also influence the approach around the use of the mouse model to test specific pathways and therapeutic approaches in atherosclerosis. In addition, the computational approach is robust and could (and likely will) be applied to a variety of other common complex conditions.
**Reviewer #2 (Public Review):**
Summary:Mouse models are widely used to determine key molecular mechanisms of atherosclerosis, the underlying pathology that leads to coronary artery disease. The authors use various systems biology approaches, namely co-expression and Bayesian Network analysis, as well as key driver analysis, to identify co-regulated genes and pathways involved in human and mouse atherosclerosis in artery and liver tissues. They identify species-specific and tissue-specific pathways enriched for the genetic association signals obtained in genome-wide association studies of human and mouse cohorts.Strengths:The manuscript is well executed with appropriate analysis methods. It also provides a compelling series of results regarding mouse and human atherosclerosis.Weaknesses:The manuscript has several weaknesses that should be acknowledged in the discussion. First, there are numerous models of mouse atherosclerosis; however, the HMDP atherosclerosis study uses only one model of mouse atherosclerosis, namely hyperlipidemic mice, due to the transgenic expression of human apolipoprotein ELeiden (APOE-Leiden) and human cholesteryl ester transfer protein (CETP). Therefore, when drawing general conclusions about mouse pathways not being identified in humans, caution is warranted. Other models of mouse atherosclerosis may be able to capture different aspects of human atherosclerosis.

We appreciate the reviewer’s valuable insight! Indeed, the specific HMDP atherosclerosis model may miss important mouse pathways that could have overlapped with the human pathways. We have added this important point to the limitations section under the discussion to caution the interpretation of the human-specific pathways, as they could be present in mice but are missed by the specific HMDP atherosclerosis dataset used.

Second, the mouse aorta tissue is atherosclerotic, whereas the atherosclerosis status of the GTEX aorta tissues is not known. Therefore, it is possible that some of the human or mouse-specific gene modules/pathways may be due to the difference in the disease status of the tissues from which the gene expression is obtained.

We agree with the reviewer that GTEx vascular tissues have unclear atherosclerosis status. However, in addition to GTEx, we also included the human STARNET dataset which contains vascular tissues from human patients with CAD. Therefore, we believe the comparability of the human and mouse vascular tissue datasets is reasonable.

Third, it is unclear how the sex of the mice (all female in the HMDP atherosclerosis study and all male in the baseline HMDP study) and the sex of the human donors affected the results. Did the authors regress out the influence of sex on gene expression in the human data before performing the co-expression and preservation studies? If not, this should be acknowledged.

We acknowledge that the effect of sex in the mouse and human datasets were not regressed out in our analysis. We have added this under the limitations section.

Fourth, some of the results are unexpected, and these should be discussed. For example, the authors identify that the leukocyte transendothelial migration pathway and PDGF signaling pathway are human-specific in their vascular tissue analysis. These pathways have been extensively described in mouse studies. Why do the authors think they identified these pathways as human-specific? Similarly, the authors identified gluconeogenesis and branched-chain amino acid catabolism as human and mouseshared modules in the vascular tissue. Is there evidence of the involvement of these pathways in atherosclerosis in vascular cells?

We agree with the reviewer that these unexpected findings warrant further discussion. As pointed out by this reviewer, it is possible that the mouse HMDP atherosclerosis dataset cannot fully represent all mouse atherosclerosis biology and therefore missed the leukocyte migration and PDGF pathways that were identified in the human datasets. Regarding the surprising findings of pathways such as BCAA catabolism in vascular tissues, we acknowledge that future studies will need to further investigate such pathway predictions but also highlight that these pathway terms have many shared genes with more commonly known pathways such as the TCA cycle, which may indicate the involvement of energy metabolism in vascular tissues in CAD development. We have added these points to the discussion section under limitations and concluding remarks.

Overall, acknowledging these drawbacks and adding points to the discussion will strengthen the manuscript.
**Reviewer #1 (Recommendations For The Authors):**
1. Could the authors comment on why MEGENA produces so many more co-expression modules per tissue than WCGNA?

As described in the methods section, MEGENA uses a multi-scale clustering structure to generate network modules at different scales, with each scale representing a different compactness level of the modules. At lower compactness scales larger modules are generated; at higher compactness scales, smaller modules are generated. By using all modules obtained from different scales, the total number of modules is much larger than WGCNA which only generates a network at one scale.

1. Much of the results section involves repeating in the text lists of pathways, modules, and genes that are also listed in Figures 2 and 3. The text in this part of the results could be used more productively to focus on illustrative examples or potential new biology.

We have revised the results section to reduce repeating long lists of pathways, modules, and genes as suggested.

**Reviewer #2 (Recommendations For The Authors):**
In addition to the weaknesses I mentioned in the public review comments, there are a few minor issues that I outline below:1. The authors should introduce atherosclerosis as the underlying cause of CAD in theIntroduction. In fact, I believe there are many places in the manuscript where theauthors mean atherosclerosis instead of coronary artery disease, especially when presenting and discussing mouse results since the HMDP study did not examine the coronary arteries of mice. I believe the authors should make the appropriate changes throughout the manuscript.

We have made the changes as suggested.

1. The authors state in the introduction, "For example, mice tend to develop atherosclerotic lesions in the aorta and carotids, while humans often develop lesions in coronary arteries (Ma et al., 2012)." This is not entirely correct, so this sentence should be revised. Several models of mice show coronary artery atherosclerosis development, but most researchers study lesions in larger aortas. Further, humans develop lesions throughout the arterial tree, but perhaps what the authors meant was the most consequential plaque development is in the coronary arteries. Please rephrase.

We have rephrased the statement as suggested.

1. Last line of page 5 should read "...which will drive modules and pathways that are more likely..." not "derive"

Typo corrected.